# Evidence for Immunity against Tetanus, Diphtheria, and Pertussis through Natural Infection or Vaccination in Adult Solid Organ Transplant Recipients: A Systematic Review

**DOI:** 10.3390/microorganisms12050847

**Published:** 2024-04-24

**Authors:** Emil Lenzing, Zitta Barrella Harboe, Søren Schwartz Sørensen, Allan Rasmussen, Susanne Dam Nielsen, Omid Rezahosseini

**Affiliations:** 1Department of Infectious Diseases, Rigshospitalet, University of Copenhagen, 2100 Copenhagen, Denmark; emil.lenzing.01@regionh.dk (E.L.); zitta.barrella.harboe@regionh.dk (Z.B.H.); susanne.dam.poulsen@regionh.dk (S.D.N.); 2Department of Pulmonary Medicine and Infectious Diseases, Copenhagen University Hospital at Nordsjællands, 3400 Hillerød, Denmark; 3Department of Clinical Medicine, University of Copenhagen, 2200 Copenhagen, Denmark; soeren.schwartz.soerensen@regionh.dk; 4Department of Nephrology, Rigshospitalet, University of Copenhagen, 2100 Copenhagen, Denmark; 5Department of Surgical Gastroenterology and Transplantation, Copenhagen University Hospital—Rigshospitalet, 2100 Copenhagen, Denmark; allan.rasmussen@dadlnet.dk

**Keywords:** vaccination, infection, solid organ transplant, tetanus, diphtheria, pertussis, immunity, immunosuppression

## Abstract

(1) Background: We aim to systematically review the current evidence on immunity against tetanus, diphtheria, and pertussis in adult solid organ transplantation (SOT) recipients, either through natural infection or vaccination. (2) Methods: This systematic review was conducted per PRISMA guidelines. We assessed the risk of bias using the Cochrane RoB 2 and ROBINS-I and summarized the findings narratively due to the heterogeneity of the studies. (3) Results: Of the 315 screened articles, 11 were included. Tetanus immunity varied between 55% and 86%, diphtheria immunity from 23% to 75%, and pertussis immunity was between 46% and 82%. Post-vaccination immunity showed variation across the studies, with some indicating reductions and others no change, with antibody responses influenced by transplanted organs, gender, age, and immunosuppressive regimens. The single randomized study exhibited a low risk of bias, while of the ten non-randomized studies, six showed moderate and four serious risks of bias, necessitating cautious interpretation of results. (4) Conclusions: SOT recipients exhibit considerable immunity against tetanus and diphtheria at transplantation, but this immunity decreases over time. Although vaccination can enhance this immunity, the response may be suboptimal, and the increased antibody levels may not persist, underscoring the need for tailored vaccination strategies in this vulnerable population.

## 1. Introduction

Solid organ transplantation (SOT) recipients require lifelong immunosuppressive therapy and, therefore, are at a higher risk of infections compared to the general population [1,2]. A proportion of the infections in SOT recipients are vaccine-preventable, and almost 12% of SOT recipients experience at least one episode of a vaccine-preventable infection (VPI) within 12 years post-transplantation [2].

Tetanus, diphtheria, and pertussis are rare bacterial VPIs in adult SOT recipients [3,4,5,6]. In a recent cohort of SOT recipients from Switzerland, no case of diphtheria or tetanus was found within 12 years post-transplantation; however, two cases of pertussis were detected, resulting in an incidence rate of 10 per 100,000 person-years of follow-up [2]. Despite the low incidence of tetanus, diphtheria, and pertussis infections, treating these infections in SOT recipients is challenging and requires specialized intensive care units with high mortality [3]. A case report from Brazil documented a patient acquiring tetanus six years after renal transplantation, leading to acute kidney injury and a 37-day hospital stay [3]. In France, another case involved a renal transplant recipient who developed generalized tetanus 12 years post-transplantation; although anti-tetanus antibodies were detectable, the patient required prolonged intensive care and could not ingest food for 11 days due to trismus [4]. Additional reports from the USA and Spain have described pertussis infections in renal transplant recipients that were diagnosed and treated after significant delays, with patients experiencing a month of coughing [5,6]. Due to the potential risk of severe infection and poor outcomes, vaccination with tetanus and diphtheria toxoids vaccine (Td) or tetanus, diphtheria, and acellular pertussis (Tdap) vaccine are recommended in adult SOT recipients before or after transplantation [7,8]. However, the recommendations primarily rely on studies conducted on children, and data on adult SOT recipients are scarce [7,8].

Furthermore, while lifelong immunosuppressive therapy affects humoral and cellular immune responses and the antibody decay rates may vary in SOT recipients, evaluation for serologic response following vaccination against tetanus, diphtheria, and pertussis has not been universally recommended post-transplantation [7,8,9,10]. Therefore, there is a need to systematically review the literature on immunity against tetanus, diphtheria, and pertussis in adult solid organ transplant recipients. Such information can help to fill gaps in knowledge, point to need for further research, and aid in planning preventive strategies in case of an epidemic.

Therefore, our aim is to systematically review the literature regarding immunity via natural infection or vaccination against tetanus, diphtheria, and pertussis in adult solid organ transplant recipients.

## 2. Materials and Methods

### 2.1. Search Strategy

We adhered to the Preferred Reporting Items for Systematic Reviews and Meta-Analyses (PRISMA 2020) statement throughout this systematic review [11]. The main clinical question we addressed was: “Is there any evidence concerning immunity through natural infection or vaccination against tetanus, diphtheria, and pertussis in adult solid organ transplant recipients?”.

To formulate this question, we followed the Population, Intervention, Comparator, Outcome, and Study design (PICOS) process. To find the relevant literature, we conducted a comprehensive search in various databases, including PubMed, MEDLINE, EMBASE, Scopus, and Web of Science. The search spanned from 1 January 1950 to 10 August 2023. Two independent investigators (EL and OR) performed the initial screening of papers, utilizing predetermined search terms and evaluating titles and abstracts for potential relevance. Subsequently, relevant papers were thoroughly assessed in full text, and inclusion criteria were rigorously applied to determine their eligibility for the review. In instances where discrepancies arose during the screening and inclusion process, a third independent investigator (ZBH) was involved to resolve any disagreements and ensure consistency. We also reviewed the reference list of the included studies and conducted a manual search on Google to find any relevant studies not identified through the mentioned databases.

The study protocol for this systematic review was registered on the open science framework (OSF) [12]. OSF (https://osf.io/ (accessed on 3 March 2024)) is a free, open-source software project designed to support the research lifecycle. The main aim of OSF is to facilitate open collaboration in scientific research by providing a platform. Researchers can conduct, manage, and share their work more transparently via OSF [12]. By following these systematic procedures, we aimed to maintain the integrity and transparency of our research process.

### 2.2. Inclusion Criteria

We included studies on vaccination against diphtheria, tetanus, and pertussis in recipients of solid organ transplants (SOTs) aged 18 years or older, containing information about humoral or cellular immune response. We included observational studies, randomized controlled trials, and non-randomized controlled trials while excluding reviews, case series, case reports, ideas, editorials, and opinions. Furthermore, only studies published in English were considered for inclusion.

### 2.3. Full Search Strategy

To execute our search strategy, we employed a combination of MeSH terms and performed a separate search for free-text terms in the PubMed/MEDLINE databases. The following MeSH term combination was used in PubMed, resulting in 32 hits: ((“Diphtheria”[Mesh] OR “Bordetella pertussis”[Mesh] OR “Tetanus”[Mesh] OR “Tetanus Toxoid”[Mesh]) AND (“Organ Transplantation”[Mesh] OR “Transplants”[Mesh]) AND (“Immunity”[Mesh] OR “Vaccination”[Mesh])).

For the free-text search, we used the following combination of search terms and found 207 hits in PubMed: (((((diphtheria) OR (Corynebacterium diphtheriae)) OR (diphtheria toxin)) OR (((tetanus) OR (tetanus toxin)) OR (clostridium tetani))) OR ((pertussis) OR (Bordetella pertussis))) AND ((vaccine) OR (vaccination) OR (Immunity) OR (Antibody) OR (Immunization)) AND (organ transplant). The same combination of free-text terms was used to search EMBASE, Scopus, and Web of Science. 

All search results were imported into Covidence [13]. Covidence (https://www.covidence.org/ (accessed on 3 March 2024)) is a web-based software tool that helps to facilitate the process of systematic reviews. Various stages of the systematic review, including screening search results, removing duplicate studies, selecting studies for inclusion, extracting data, and assessing the risk of bias, can be performed via Covidence. Importantly, the tool supports collaboration among team members, allowing for transparent and coordinated review decisions [13].

### 2.4. Data Extraction and Risk of Bias 

We extracted data using Extraction 2.0 in the Covidence online platform [13]. A project is introduced to the platform, and investigators are invited via email. The investigators can import search results from different databases into the project and the platform helps to remove duplicate studies found across several databases. Each investigator can independently screen the imported studies by title and abstract; if a study seems relevant, they can include it for full-text review or exclude it, providing a reason. It is possible to design data extraction forms and extract data directly within the platform. Additionally, Covidence supports communication and task management among team members. Both data on humoral and cellular immune responses were extracted and recorded in the data extraction forms. We assessed the risk of bias using the Cochrane Risk of Bias 2 for randomized clinical trials (RoB 2) [14] and the Risk of Bias in Non-randomized Studies of Interventions (ROBINS-I) [15]. The risk-of-bias visualization tool (Robvis) was used to visualize the quality of the included studies with traffic light plots [16]. Risk-of-bias tools are standard instruments designed to assess the quality and integrity of the studies. The tools utilize a set of standard signaling questions that follow a predefined algorithm to systematically uncover any elements within a study that may introduce bias. Risk-of-bias tools evaluate various aspects of the study design, such as the methodology, participant selection, data collection processes, reported results, and statistical analysis techniques. Therefore, risk-of-bias tools help to identify potential weaknesses or areas of concern that could compromise the validity and reliability of the studies. The final goal of using risk-of-bias tools is to ensure that the results and conclusions that are drawn from clinical research are based on solid, unbiased evidence, thereby enhancing the utility of the findings [14,15]. The judgment for the risk of bias was categorized as either low, some concerns, or high, and the judgment for non-randomized clinical trials was grouped as low, moderate, or high [14,15]. Due to the included studies’ heterogeneity, a meta-analysis could not be performed, and the data were summarized narratively.

## 3. Results

We screened 315 and included 11 articles, comprising 10 non-randomized studies, and 1 clinical randomized study (Figure 1). The only randomized study was a sub-study of a multicenter trial that investigated the humoral and cellular immune responses to tetanus vaccines in kidney transplant recipients undergoing immunosuppression [17]. Among the non-randomized studies, ten focused on tetanus, eight on diphtheria, and only one study investigated pertussis vaccination. Characteristics, definitions, and limitations of the included studies are shown in Table 1.

### 3.1. What Is the Percentage of Adult SOT Recipients Who Are Immune to Tetanus, Diphtheria, and Pertussis?

#### 3.1.1. Tetanus 

Among the studies examining tetanus immunity (Table 1), Broeders et al. demonstrated that 71% of kidney transplant recipients had immunity to tetanus at the time of transplantation, which decreased to 55% one year after transplantation [10]. For comparison, the immunity to tetanus was 98% among healthy controls, who were selected from healthcare personnel [10]. In the study conducted by Krüger et al., it was discovered that 80% of the hemodialysis patients who had been vaccinated and subsequently received a kidney transplant within an unspecified period in the first year exhibited immunity to tetanus. However, this percentage declined to 60% among those who underwent a kidney transplant at an unspecified time within five years after vaccination [9]. A comparison of lung transplant recipients and healthy individuals by Rohde et al. revealed that with a median of 5 years after transplantation, 85% of transplant recipients were immune to tetanus; however, the proportion was 100% in healthy individuals [18]. Chesi et al. reported that, after a median of 67 months post-transplant, 85% of liver transplant recipients were found to be immune to tetanus [19]. On the other hand, kidney transplant recipients showed a slightly higher proportion of 86% immunity to tetanus, at an earlier median post-transplant duration of 46 months [19]. Boey et al. showed that with a median of seven years after transplantation, 80% of heart and lung transplant recipients were immune to tetanus [20].

**Table 1 microorganisms-12-00847-t001:** Characteristics, definitions, and limitations of the included studies.

First Author/Publication Year/Country in Which the Study Was Conducted	Study Design	Population Description	Antibody Cut-Offs	Limitations
Boey/2021/Belgium [20]	Cross-sectional study	Six groups of patients with chronic diseases (1052 patients), including 230 heart or lung transplant recipients, were investigated. **SOT recipients** Female: 73/230 (32%) Median age (range): 59 (19–87) years Median (range) time from transplantation: 7 (1–29) years	**Diphtheria** Seronegative: Anti-DT < 0.01 IU/mL Seroprotective: Anti-DT ≥ 0.1 IU/mL **Tetanus** Seronegative: Anti-TT titers < 0.01 IU/mL Seroprotective: Anti-TT titers ≥ 0.1 IU/mL **Pertussis** Seropositive: Anti-PT, anti-FHA (filamentous hemagglutinin), and anti-Prn (pertactin) titers ≥ 5 IU/mL. Pertussis infection or vaccination in the past two years: Anti-PT titers ≥ 50 IU/mL Recent infection or vaccination: Anti-PT titers ≥ 100 IU/mL	Single-center study; lack of documented vaccination history for all the patients
Blanchard-Rohner/2019/Switzerland [21]	Cross-sectional study	This study included two groups of transplant recipients, and anti-TT antibodies were measured on transplantation day. **Group 1:** Sixty-five (29 liver (±kidney), 25 kidney, one lung, four heart, and six pancreas/Langerhans islets) SOT recipients who were transplanted during 2013 and before the implementation of a systematic vaccination approach. Female: 21/65 (32%) Median (IQR) age: 53 (46–61) **Group 2:** A systematic vaccination approach was introduced in 2014, and 219 SOT candidates were included from January 2014 to November 2015. Fifty-four (27 Liver (±kidney), 11 kidney, eight lung, six heart, and two pancreas/Langerhans islets) out of 219 were transplanted during the study. Female: 14/54 (26%) Median (IQR) age, years: 56 (46–63)	**Tetanus** A vaccine was offered if Anti–TT < 500 IU/L (<0.5 IU/mL) Seroprotective: Anti–TT titers > 100 IU/L L (<0.1 IU/mL)	Lack of documented vaccination history for all the SOT recipients; lack of follow-ups after transplantation
Rohde/2014/United States [18]	Cross-sectional study	Seventy-five lung transplant recipients and 36 healthy individuals were included. Serum samples were collected from 2004 to 2008. **Lung transplant recipients:** Female: 41/75 (55%) Median (IQR) age, years: 57 (50–65) * Median (range) time from transplantation: 5.3 (0.17–16.6) years **Controls:** Female: 17/36 (47%) Median (IQR) age, years: 46 (37–53) *	**Diphtheria** Seroprotective: Anti-DT ≥ 0.1 IU/mL **Tetanus** Seroprotective: Anti-TT titers ≥ 0.15 IU/mL	Only the most recent Td or Tdap vaccination is reported, not any prior vaccinations. Lack of documented vaccination history for last Td or Tdap for 20% of the SOT recipients.; lack of regular follow-ups after transplantation
Broeders/2013/Australia [10]	Cohort study	Ninety-four kidney transplant recipients with a functional graft and 49 healthy hospital workers were included in this study. Anti-TT antibodies were measured on transplantation day and one year later. **Kidney transplant recipients:** Female: 40/94 (43%) Median (IQR) age, years: 46 (33–59) ** **Controls:** Female: 24/49 (49%) Median (IQR) age, years: 42 (32–52) **	**Tetanus** Seroprotective: Anti-TT titers ≥ 0.1 IU/mL	No information about previous vaccination
Puissant-Lubrano/2010/France [22]	Cross-sectional study	The immune response to tetanus vaccination was investigated before and one month after vaccination in 39 kidney transplant recipients who received different immunosuppressive agents, including antiproliferative agents and/or calcineurin inhibitors plus steroids. **Group 1:** Thirteen out of the thirty-nine kidney transplant recipients received rituximab with a median (IQR) of 9 (4–11.5) months before vaccination. Female: 3/13 (23%) Median (IQR) age, years: 55 (40–64) Median (range) time from transplantation, years: 6.9 (2.5–15.3) *** **Group 2:** Twenty-six out of the thirty-nine kidney transplant recipients did not receive rituximab. Female: 11/26 (42%) Median (IQR) age, years: 48 (36–59) Median (range) time from transplantation, years: 2.3 (2–4) *** **Controls:** Serum specimens from 30 healthy blood donors were used to compare the antibody response before vaccination.	**Diphtheria** Seroprotective: Anti-DT ≥ 0.1 IU/mL **Tetanus** Seroprotective: Anti-TT titers ≥ 0.15 IU/mL A 4-fold increase in anti-TT after vaccination was considered as significant response to vaccination.	Lack of detailed vaccination records
Struijk/2010/The Netherlands [17]	Randomized controlled trial	This study was a sub-study of an open, randomized, multicenter trial. Thirty-six stable kidney transplant recipients and 13 age- and sex-matched healthy persons were included as controls. Kidney transplant recipients were assigned into 3 groups (12 per group). Kidney transplant recipients were not vaccinated against TT in the previous five years, and were within the 2nd year after transplantation and received double immunosuppressive maintenance therapy consisting of prednisolone with CsA, MPA, or everolimus from 6 months after transplantation. The controls were excluded if they received immunosuppressants or were vaccinated against tetanus within the previous five years. **Group 1 (Prednisolone + CsA)** Female: 2/12 (17%) Median (IQR) age, years: 58 (34–72) Median (range) time from transplantation, years: **Group 2: (Prednisolone + MPA)** Female: 3/12 (25%) Median (IQR) age, years: 60 (30–70) Median (range) time from transplantation, years: **Group 3: (Prednisolone + everolimus)** Female: 4/12 (33%) Median (IQR) age, years: 50 (27–68) **Controls:** Female: 5/13 (38%) Median (IQR) age, years: 55 (42–63)	Participants were vaccinated with three vaccines simultaneously. Immunocyanin (Immucothel, Biosyn Arzneimittel GmbH, Fellbach, Germany), Tetanus toxoid (Aventis Pasteur MSD Brussels, Belgium), and Polyvalent Pneumococcal vaccine (Pneumovax, Merck Sharp and Dohme, Haarlem, The Netherlands). Blood specimens were collected before and 14 days after vaccination. Anti-TT antibody concentrations were measured using ELISA, and TT-specific cellular responses were measured using ELISPOT assay.	
Chesi/2009/Germany [19]	Cross-sectional study	Four hundred sixty-four adult SOT recipients (267 liver and 197 kidney transplants) were included. **Liver transplant recipients:** Female: 112/267 (42%) Median (IQR) age, years: 57 (20–79) Median (range) time from transplantation, years: 5.6 (0.5–20) Immunosuppressive therapy: 51%, 42%, and 7.1% received one, two, and three or more immunosuppressive drugs **Kidney transplant recipients** Female: 97/197 (49%) Median (IQR) age, years: 51 (18–79) Median (range) time from transplantation, years: 3.8 (0.5–31) Kidney transplant recipients were younger, with a shorter time from transplantation (*p* < 0.005). Immunosuppressive therapy: 0.5%, 35%, and 65% received one, two, and three or more immunosuppressive drugs.	Anti-TT and anti-DT IgG antibodies were measured by ELISA (Virion Serion, Wurzburg, Germany) on a Behring ELISA Processor, BEP III. **Diphtheria** Seroprotective: Anti-DT ≥ 0.1 IU/mL **Tetanus** Seroprotective: Anti-TT titers ≥ 0.1 IU/mL	Patients reported vaccination status; only 159 patients (34.3%) possessed a vaccination certificate, more frequently kidney transplant recipients than liver transplant recipients (43.5% vs. 27.3%; *p* < 0.005). Immunosuppressive therapy is only reported as 1, 2, or 3 or more drugs.
Goldfarb/2001/United States [23]	Cohort study	Sixty-seven lung transplant recipients were grouped by IgG level. **Low IgG** Female: 11/25 (44%) Median (IQR) age: 50 (21–60) **Moderately low IgG** Female 11/22 (50%) Median (IQR) age: 51 (17–61) **Normal IgG** Female 11/20 (55%) Median (IQR) age: 37 (11–59)	Hypogammaglobulinemia was defined as an IgG level of <600 mg/dL. Low IgG levels were defined as less than 400 mg/dL. Moderately low IgG was defined as levels between 400 and 600 mg/dL. Normal IgG levels were above 600 mg/dL.	Single-center study; only 59 of 130 on-site transplants followed in the study; different pre and post-transplant amount of participants in humoral immune surveys
Krüger/2001/Germany [9]	Cohort study	Seventy-one anti-TT and anti-DT seronegative patients on hemodialysis were vaccinated simultaneously against tetanus and diphtheria and followed for five years. Anti-TT and anti-DT antibodies were measured one and five years after vaccination. Thirty-five patients were transplanted within five years of follow-up, and antibody concentrations were available for fifteen alive kidney transplant recipients in the fifth year. The kidney transplant recipients’ characteristics have not been reported. No re-vaccination was performed within the last five years.	**Diphtheria** Seroprotective: Anti-DT ≥ 0.1 IU/mL **Tetanus** Seroprotective: Anti-TT titers ≥ 0.1 IU/mL	Loss of follow-up for a considerable proportion of the transplant recipients: the kidney transplant recipients’ characteristics were not reported.
Huzly/1997/Germany [24]	Cohort study	One hundred sixty-four kidney transplant recipients and 106 healthy volunteers were included. **Kidney transplant recipients** Female: 58/164 (59.9%) Median (IQR) age: 43 (16–66) Median time sice renal transplant (IQR) yr: 2 (1–24) Immunosuppressive therapy: 83 (50.6%) received cyclosporine, azathioprine and prednisone. Twenty-nine (17.7%) received azathioprine and prednisone. Twenty-seven (16.4%) received cyclosporine and prednisone. Sixteen (9.8%) received cyclosporine and azathioprine. Nine (5.5%) received only cyclosporine **Controls:** Female: 67/106 (63.2%) Median (IQR) age: 42 (18–68)	**Diphtheria** Relative protective: Anti-DT ≥ 0.01 IU/mL Seroprotective: Anti-DT ≥ 0.1 IU/mL **Tetanus** Seroprotective: Anti-TT titers ≥ 0.01 IU/mL	Only 55 of 164 patients were part of the 12-month follow-up.
Girndt/1995/Germany [25]	Cohort study	Fifty-seven anti-TT seronegative patients with chronic kidney disease, including seven kidney transplant recipients and fifteen controls from the outpatient hypertension clinic **Kidney transplant recipients** Female: 0/7 (0%) Mean (SD) age: 47.8 (8.4) Median time since kidney transplantation (IQR), years: 2 (1–24) Mean (SD) serum creatinine (mg/dL): 1.39 (0.30) All received prednisolone, cyclosporine, and azathioprine as immunosuppressive treatment. **Controls** Female: 8/15 (0%) Mean (SD) age: 51.3 (13.3) Mean (SD) serum creatinine (mg/dL): 0.97 (0.25)	**Tetanus** Seroprotective: Anti-TT titers ≥ 0.01 IU/mL	A limited number of participants; only patients vaccinated for tetanus more than ten years ago were tested for seronegativity.

* Controls were younger than lung transplant recipients (*p* = 0.0001); ** Controls were younger (*p* = 0.039); *** Transplant recipients in Group 2 had a shorter median time from transplantation (*p* = 0.01).

#### 3.1.2. Diphtheria

In the study by Boey et al., only 23% of heart and lung transplant recipients were immune to diphtheria at a median of seven years post-transplantation [20]. In the study conducted by Krüger et al., 52% of hemodialysis patients who were vaccinated and subsequently received a kidney transplant at an unspecified time within the first year showed immunity to diphtheria. However, this proportion decreased to 40% among those who had undergone a kidney transplant at an unspecified time within five years after vaccination [9]. The study by Rohde et al. revealed that a lower proportion of lung transplant recipients than healthy individuals were immune to diphtheria (75% vs. 94%) [18]. In the study by Chesi et al., 73% of liver transplant recipients and 60% of kidney transplant recipients were immune to diphtheria [19].

#### 3.1.3. Pertussis

The immunity to pertussis varied among heart and lung transplant recipients, depending on the antigen: 46% were immune to pertussis toxin, 82% to filamentous hemagglutinin, and 58% to pertactin, measured at a median (range) of seven (1–29) years post-transplantation [20]. 

### 3.2. Do Adult SOT Recipients Elicit an Antibody Response When Vaccinated against DTP?

Blanchard-Rohner et al. implemented a systemic vaccination program for SOT candidates and demonstrated that catch-up immunizations notably enhanced immunity against tetanus among adult SOT recipients. The percentage of individuals with immunity (IgG > 500 IU/L) increased from 77% to 91% post-vaccination with a median of 4.6 months of follow-up (*p* = 0.03) [21].

Additionally, Girndt et al. found that 71% of kidney transplant recipients seroconverted (0.01 > IU/mL) after one tetanus toxoid vaccination, and 85% seroconverted after two and three subsequent doses, which were administered 4 and 24 weeks after, respectively [25]. These results were not significantly different from controls [25]. After three vaccinations, the antitetanus toxin (anti-TT) antibody concentration was 0.81 +/− 0.66 IU/mL in kidney transplant recipients and 2.44 +/− 1.32 IU/mL in controls (*p* < 0.05) [25].

Regarding diphtheria, Krüger et al. investigated twelve kidney transplant recipients who had not been previously vaccinated. One year after vaccination, they found a substantial increase in diphtheria antibody levels, from 0.06 +/− 0.03 IU/mL to 0.33 +/− 0.52 IU/mL [9].

Furthermore, Huzly et al. compared antibody responses between transplant recipients and healthy controls, and the mean antibody increase was lower in transplanted patients for both tetanus (2.23 vs. 4.5 IU/mL) and diphtheria (0.76 vs. 1.74 IU/mL) [24]. The antibody values were measured before and four weeks after vaccination [24].

### 3.3. What Is the Antibody Decay Profile after Vaccination against DTP in Adult SOT Recipients?

In kidney transplant recipients, anti-TT antibody concentrations exhibited a significant decrease by a factor of four within the first year post-transplantation (*p* < 0.001). The median half-life of anti-TT antibodies was determined to be approximately 7.7 months [10]. Among lung transplant recipients and healthy controls, median anti-TT antibody concentrations were 1.3 IU/mL and 3.2 IU/mL, respectively (*p* = 0.01) [18]. Furthermore, time elapsed since the last Td or Tdap vaccination did not correlate with anti-TT antibody concentrations in lung transplant recipients (r= −0.087, *p* = 0.52) but did show a correlation in healthy individuals (r= −0.48, *p* = 0.009) [18]. A study involving seronegative kidney transplant recipients revealed that anti-TT antibody concentrations increased post-vaccination, with mean levels of 0.04 ± 0.01 IU/mL at baseline, 0.68 ± 1.25 IU/mL at one year, and 0.78 ± 1.48 IU/mL at the fifth year after vaccination [9]. 

Comparatively, in lung transplant recipients and healthy controls vaccinated less than five years before serum collection, median anti-diphtheria (anti-DT) antibody concentrations were notably lower in lung transplant recipients, with levels of 0.26 IU/mL compared to 0.89 IU/mL in healthy individuals (*p* = 0.001) [18]. Similar to the trend observed with anti-TT antibodies, time since the last Td or Tdap vaccination did not correlate with anti-DT antibody concentrations in lung transplant recipients (r = 0.14, *p* = 0.3) but showed a correlation in healthy individuals (r = −0.53, *p* = 0.004) [18]. Furthermore, the study by Krüger et al. highlighted a significant decrease in diphtheria antibody levels, from 0.33 +/− 0.52 IU/mL one year after vaccination to 0.14 +/− 0.23 IU/mL five years after vaccination [9].

### 3.4. Gender and Age Differences in Antibody Response

In the study by Rohde et al., median anti-TT antibody concentrations were comparable between female and male lung transplant recipients, with values of 1.7 IU/mL and 1.9 IU/mL, respectively (*p* = 0.93) [18]. Furthermore, a negative correlation was observed between recipient age and both anti-TT and anti-DT titers (r = −0.34, *p* = 0.003) [18]. The study by Chesi et al. highlighted that anti-TT titers were negatively correlated with female gender (*p* < 0.001) and age (*p* < 0.05) [19], although Krüger et al. did not find any significant correlation between sex and vaccination response in kidney transplant recipients [9].

Rohde et al. reported that the median anti-DT antibody concentrations did not differ significantly between female and male participants, with values of 0.25 IU/mL and 0.50 IU/mL, respectively (*p* = 0.24) [18]. However, female transplant recipients exhibited a higher likelihood of being anti-DT seronegative compared to males (26% vs. 9%, *p* = 0.015) [18]. The study by Goldfarb et al. suggested a trend in inverse correlation between the mean age of lung transplant recipients and anti-DT IgG levels, with the lowest IgG group having an average age of 45 years (under 400 mg/dL IgG), the moderately low group with an age of 48 years (400–600 mg/dL IgG), and the normal group with an age of 37 years (above 600 mg/dL) [23]. Huzly et al. showed, in a multivariable analysis, that antibody response to vaccination was not correlated with age or gender [24]. 

### 3.5. Is There a Difference in the Antibody Response When Vaccine Boosters Are Administered before versus after Transplantation?

In the study by Rohde et al. involving lung transplant recipients, two distinct groups were examined [18]. The first group, comprising 33 recipients, had received their last Td or Tdap booster at a median of 28 months before transplantation. The second group, consisting of 25 recipients, received the last booster vaccine dose at a median of 38 months post-transplantation. The antitetanus toxoid (anti-TT) antibody concentrations were measured on a median of 98 and 45 months after the last vaccination in the first and second groups, respectively. Notably, the anti-TT antibody concentration was found to be 1.5 IU/mL in the first group and 1.6 IU/mL in the second group, with no statistically significant difference between the groups (*p* > 0.1). Seropositivity against tetanus was observed in 94% of the first group and 76% of the second group (*p* = 0.12) [18]. The lung transplant recipients who had received diphtheria immunization exhibited an anti-diphtheria toxoid (anti-DT) antibody concentration of 0.41 IU/mL. In comparison, the second group of 25 recipients who received a post-transplant booster vaccine demonstrated an anti-DT antibody concentration of 0.17 IU/mL. However, the differences in anti-DT antibody concentrations between the groups were not statistically significant (*p* > 0.1). Seropositivity rates for diphtheria were 85% in the first group and 64% in the second group (*p* = 0.13) [18].

### 3.6. Immunosuppressive Combinations and Immune Response to Vaccination

Puissant-Lubrano et al. conducted a study involving kidney transplant recipients to investigate the response to tetanus toxoid vaccination in patients who receive rituximab [22]. A responder was defined as a fourfold increase in anti-TT antibodies one month after vaccination [22]. Among the recipients who received rituximab, 31% (4 out of 13) were responders, while among those who did not receive rituximab, 61% (16 out of 26) were responders (*p* = 0.096) [22]. All the kidney transplant recipients received immunosuppressive agents, including antiproliferative agents and/or calcineurin inhibitors, plus steroids. The study concluded that rituximab hinders the secondary immune response following tetanus toxoid vaccination, although it does not completely eliminate it in every patient [22]. The study by Goldfarb et al. investigated lung transplant recipients who received combination immunosuppressive therapy and had hypogammaglobulinemia (IgG < 600 mg/dL) post-transplantation [23]. Goldfarb et al. found that 19% and 15% of the lung transplant recipients had low antibody titers against tetanus and diphtheria, respectively [23].

In a clinical trial by Struijk et al., the response to tetanus vaccination was assessed in kidney transplant recipients and healthy controls [17] The recipients were divided into three treatment arms: arm 1 received prednisolone plus cyclosporine, arm 2 received prednisolone plus mycophenolate, and arm 3 received prednisolone plus everolimus. Anti-TT antibodies were measured before and fourteen days after vaccination. Fourteen days after vaccination, a significant rise in antitetanus antibody concentration was observed in healthy controls, arm 1, and arm 3 but not in arm 2 [17]. The increase was significantly lower in arm 1 and arm 2 compared to healthy controls, while no significant difference was noted between arm 2 and arm 3. Additionally, the study found an increase in IL-2-, IFN-gamma-, and IL-4-producing peripheral blood mononuclear cells (PBMCs) in response to TT stimulation. IL-2-producing PBMCs were significantly elevated in arm 1, arm 3, and controls, while IFN-gamma-producing PBMCs showed a significant increase only in arm 1 and controls. The number of IL-4-producing PBMCs increased significantly in all transplant recipient arms and controls [17].

The study by Broeders et al. discovered that the levels of tacrolimus and CsA at both 3 and 6 months after vaccination were not significantly different among renal transplant recipients who lost detectable anti-TT antibodies 12 months after vaccination [10]. When comparing the induction treatments of ATG, basiliximab, or no induction treatment and their relation to the loss in detectable anti-TT titers, no significant differences between the groups were observed (*p* = 0.6) [10].

The study by Chesi et al. investigated diphtheria vaccination response in kidney transplant recipients receiving multidrug immunosuppressive therapy compared to those receiving a single immunosuppressive drug [19]. The recipients on multidrug therapy exhibited significantly lower anti-DT antibody levels [19]. Contrary to the aforementioned study, the study by Huzly et al. did not find an influence of different immunosuppressive medications on the immune response to diphtheria or tetanus vaccines. The authors speculated that these vaccines likely acted as “recall” antigens, indicating established immunological memory prior to the initiation of immunosuppressive treatment [24].

### 3.7. Risk of Bias

The overall risk of bias was low in the single included randomized study by Struijk et al. [17] (Figure 2). 

Among the ten non-randomized studies, none had a low risk of bias; six had a moderate risk of bias; and four had a serious risk of bias (Figure 3). The two domains that caused the serious risk of bias were due to the selection of participants and the risk of bias due to missing data. All the non-randomized studies had a moderate risk of bias due to confounding, which is expected in most non-randomized studies.

## 4. Discussion

Our systematic review has revealed a noteworthy variation in seroprotection against tetanus, diphtheria, and pertussis among SOT recipients across different studies, both pre- and post-transplantation. In evaluating the risk of bias in these studies, the only randomized study exhibited a low risk of bias; however, none of the ten non-randomized studies could be categorized as low risk of bias. 

Although it varies by geographical location and in different populations, the estimated global coverage for tetanus, diphtheria, and pertussis vaccination is approximately 83% [26], which is consistent with the observations from the studies we reviewed. For instance, in the study by Broeders et al., 71% of kidney transplant recipients had protection against tetanus measured on the day of the kidney transplant, although it declined to 55% within the first post-transplant year [10]. The study by Rohde et al. demonstrated 85% protection against tetanus and 75% against diphtheria in lung transplant recipients [18]. These findings suggest that most SOT recipients possess pre-existing immunity to tetanus, diphtheria, and pertussis antigens at the time of transplantation, likely attributable to prior vaccination or natural infection. 

There is a noticeable decline in immunity to tetanus, diphtheria, and pertussis post-transplantation [10,18,21]. However, the decline in immunity is not uniform for all components of the Td/Tdap vaccine. A study of kidney transplant recipients revealed that while nearly all transplant recipients maintained protective levels of tetanus antibodies after a year, a significant portion (38%) had diphtheria antitoxin levels below the protective threshold [24]. This variation in immunity can be attributed to several factors. Gender and age have been investigated as factors influencing immunity; however, the findings in studies that we included were inconsistent [9,18,19,24]. There appears to be a trend toward diminished immunity with age, which may be attributed to a reduction in immune function associated with aging, a process known as immunosenescence [27]. However, studies including pediatric SOT recipients have also shown a decline in immune response to diphtheria and tetanus vaccines [28,29]. Thus, factors other than age may also play a significant role. For instance, post-transplant immunosuppressive treatment can lead to reduced protective antibody levels [17,18]. A study from France indicated that lung transplant recipients receiving rituximab had a lower immune response to the tetanus vaccine than those not on rituximab [22]. Immunosuppressive agents that are used as maintenance therapy in SOT recipients, such as cyclosporine and tacrolimus, primarily affect T-cell immunity. However, the combination therapy can impact both humoral and cellular immune responses to vaccination. This underscores the importance of studying the effects of different immunosuppressive regimens on both short-term and long-term responses to vaccination.

The rate of antibody decay following vaccination plays a role in the protection against microorganisms. It was shown that while there was an initial increase in antibody titers post-vaccination, kidney transplant recipients exhibited a significant decrease in antitetanus and anti-diphtheria antibody concentrations within the first year post-transplantation [24]. Furthermore, a significant number of the kidney transplant recipients had non-protective diphtheria antitoxin levels after 12 months [24]. The observed pattern of faster antibody decay in SOT recipients highlights the need for further research into the effects of booster vaccinations in this population. 

It is important to note that while SOT recipients can mount an antibody response to booster vaccinations, evidence concerning the timing of vaccination and the effects of various immunosuppressive combinations on the immune response to vaccines support recommendations that encourage at-risk adult SOT candidates and recipients to get vaccinated pre- or post-transplantation (8). Therefore, SOT candidates and recipients should be informed about vaccination recommendations based on national or international vaccination guidelines. Implementing a systematic vaccination approach can help to increase vaccine uptake in SOT candidates and recipients [21].

While the majority of studies that were included in this systematic review focused on immunity to diphtheria and tetanus among adult SOT candidates and recipients, there was scarce information available about pertussis. The available data were primarily derived from a single-center, cross-sectional study, which showed variability in the immune response to pertussis based on the specific antigens examined [20]. The study included heart and lung transplant recipients and showed that, on average, seven years post-transplantation, more than 80% of the transplant recipients were immune against the filamentous hemagglutinin, whereas fewer than 50% were immune against pertussis toxin [20]. In line with these observations, it has been mentioned in the literature that the short-term immune response to pertussis toxin in kidney transplant recipients is suboptimal [30]. Nevertheless, the immune response to pertussis antigens in adult SOT candidates and recipients requires further research. 

In this systematic review, we adhered to the PRISMA statement and used a systematic procedure to gather the relevant literature, ensuring research integrity and transparency. However, our study had some limitations. The inclusion criteria being restricted to studies published in English may introduce a language bias, potentially omitting valuable non-English studies. Additionally, while the search was comprehensive, it may not have included all relevant information from the grey area. Furthermore, the exclusion of case series, reports, and opinions might omit valuable clinical insights. Importantly, the assessment of risk of bias could be influenced by subjectivity in interpretation, particularly for non-randomized studies. Finally, due to the heterogenicity of the studies, it was not possible to perform a meta-analysis.

## 5. Conclusions

In conclusion, the studies under review collectively suggest that adult SOT recipients generally possess considerable immunity against tetanus, diphtheria, and pertussis at the time of transplantation. However, this immunity tends to diminish after transplantation. Although vaccination can improve immunity in this population, the response may be suboptimal, and sustained, heightened antibody levels may not be maintained. Given the inherent challenges and biases in the available evidence, further research is essential to refine immunization strategies, to investigate both humoral and cellular immunity, and enhance the durability of the immune response among SOT recipients.

## Figures and Tables

**Figure 1 microorganisms-12-00847-f001:**
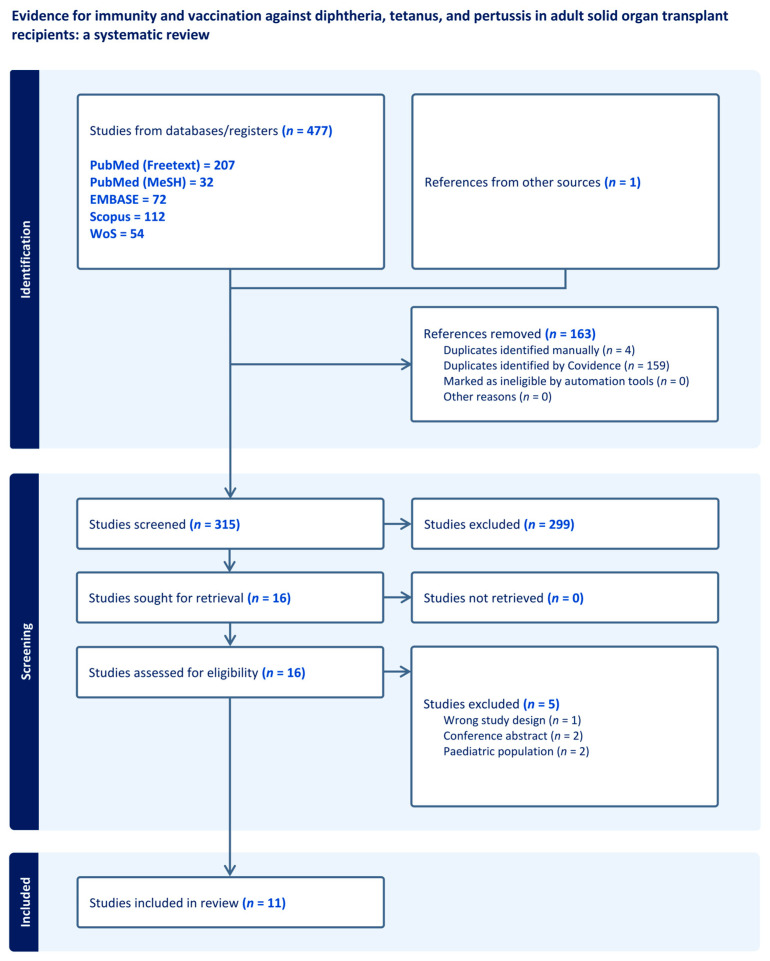
PRISMA flow diagram of the included studies.

**Figure 2 microorganisms-12-00847-f002:**
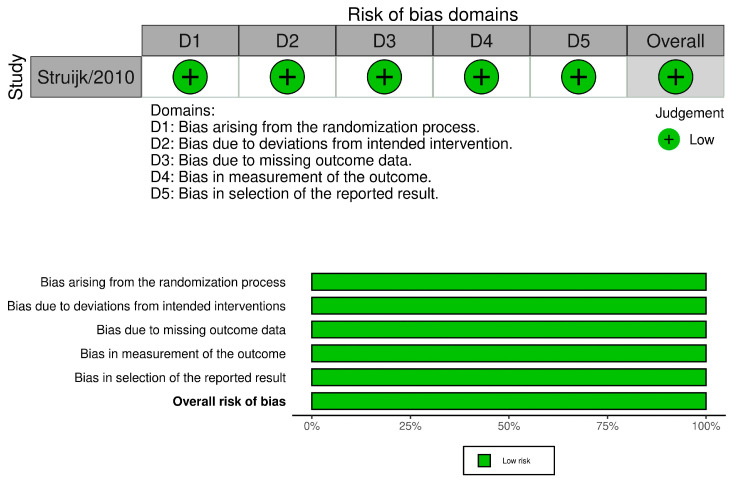
Traffic light plot and weighted bar plot for the randomized clinical trial. The overall risk of bias was low [17].

**Figure 3 microorganisms-12-00847-f003:**
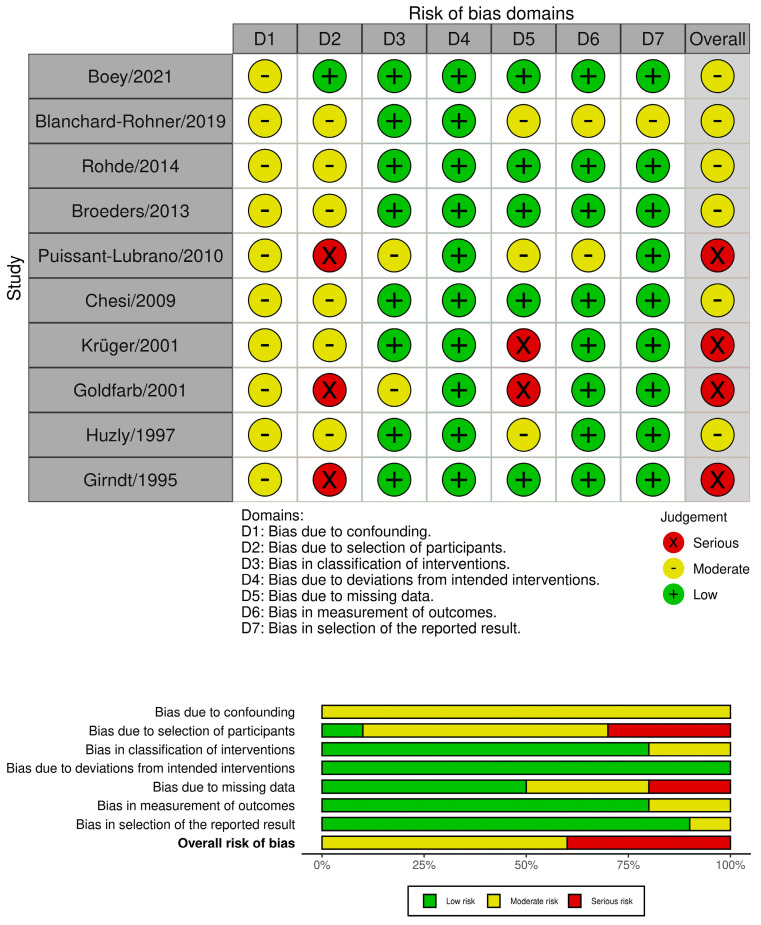
Traffic light plot and weighted bar plot for non-randomized studies. Among the 10 included non-randomized studies, there were no studies with a low risk of bias, six with a moderate risk of bias, and four with a high risk of bias. The two domains that caused the serious risk of bias were due to the selection of participants and the risk of bias due to missing data. All the non-randomized studies had a moderate risk of bias due to confounding [9,10,18,19,20,21,22,23,24,25].

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
