# Peer review of "Evidence for Immunity against Tetanus, Diphtheria, and Pertussis through Natural Infection or Vaccination in Adult Solid Organ Transplant Recipients: A Systematic Review"

_microorganisms, 2024, doi:10.3390/microorganisms12050847_

Round 1

Reviewer 1 Report

Comments and Suggestions for Authors

In general, the manuscript is well-redacted and presents an interesting revision of the literature in SOT patients regarding their immune system for three bacterial diseases. Despite that, the authors just gave a short introduction about the incidence and the importance of know-how in affecting the illnesses of Tetanus, Diphtheria, and Pertussis in this kind of patient. For that, the authors should improve the reason why this review is important to take into account for the readers. Several questions are still in doubt; for example, why is it important to know these diseases in SOT patients? Are they common? produce complex or dangerous progression? difficult treatment? Why the authors decide that Tetanus, Diphtheria and Pertussis are relevant in SOT patients? Are these bacteria resistant to antibiotics?

 On the other hand, the section Results is confusing, and the style of a review is rare. The authors just base their information on previous publications. Maybe the way to present the style of the review could be changed.

Finally, lines 390 to 397 are the same as the title conclusions (are repeated). The abbreviation SOT appears in the abstract but has not been introduced previously.

Author Response

Comments from Reviewer 1:

Comment 1, Reviewer 1:

In general, the manuscript is well-redacted and presents an interesting revision of the literature in SOT patients regarding their immune system for three bacterial diseases. Despite that, the authors just gave a short introduction about the incidence and the importance of know-how in affecting the illnesses of Tetanus, Diphtheria, and Pertussis in this kind of patient. For that, the authors should improve the reason why this review is important to take into account for the readers. Several questions are still in doubt; for example, why is it important to know these diseases in SOT patients? Are they common? produce complex or dangerous progression? difficult treatment? Why the authors decide that Tetanus, Diphtheria and Pertussis are relevant in SOT patients? Are these bacteria resistant to antibiotics?

 Response to Comment 1, Reviewer 1:

Thank you for your comment. As you suggested, we elaborated in the introduction, on page 1, line 43, to page 2, line 52, which read:

” Tetanus, diphtheria, and pertussis are rare bacterial VPIs in adult SOT recipients [3–6]. In a recent cohort of SOT recipients from Switzerland, no case of diphtheria or tetanus was found within 12 years post-transplantation; however, two cases of pertussis were detected, resulting in an incidence rate of 10 per 100,000 person-years of follow-up [2]. Despite the low incidence of the infections, vaccination with tetanus and diphtheria toxoids vaccine (Td), or tetanus, diphtheria, and acellular pertussis (Tdap) vaccine are recommended in adult SOT recipients before or after transplantation [7,8]. The recommendations are mainly due to the potential risk of severe infection and poor outcomes; however, they primarily rely on studies conducted on children, and data on adult SOT recipients are scarce [7,8].”

It now reads:

” Tetanus, diphtheria, and pertussis are rare bacterial VPIs in adult SOT recipients [3–6]. In a recent cohort of SOT recipients from Switzerland, no case of diphtheria or tetanus was found within 12 years post-transplantation; however, two cases of pertussis were detected, resulting in an incidence rate of 10 per 100,000 person-years of follow-up [2]. Despite the low incidence of tetanus, diphtheria, and pertussis infections, treating these infections in SOT recipients is challenging and requires specialized intensive care units with high mortality [3]. A case report from Brazil documented a patient acquiring tetanus six years after renal transplantation, leading to acute kidney injury and a 37-day hospital stay [3]. In France, another case involved a renal transplant recipient who developed generalized tetanus 12 years post-transplantation; although anti-tetanus antibodies were detectable, the patient required prolonged intensive care and could not ingest food for 11 days due to trismus [4]. Additional reports from the USA and Spain have described pertussis infections in renal transplant recipients that were diagnosed and treated after significant delays, with patients experiencing a month of coughing [5, 6]. Due to the potential risk of severe infection and poor outcomes, vaccination with tetanus and diphtheria toxoids vaccine (Td), or tetanus, diphtheria, and acellular pertussis (Tdap) vaccine are recommended in adult SOT recipients before or after transplantation [7,8]. However, the recommendations primarily rely on studies conducted on children, and data on adult SOT recipients are scarce [7,8].

 Comment 2, Reviewer 1:

On the other hand, the section Results is confusing, and the style of a review is rare. The authors just base their information on previous publications. Maybe the way to present the style of the review could be changed.

Response to Comment 2, Reviewer 1:

Thank you for your comment. As mentioned on page 3, lines 143-144, the studies included in this systematic review were heterogeneous, which made it impossible to combine the data for a meta-analysis. Consequently, the data were summarized narratively.

Comment 3, Reviewer 1:

Finally, lines 390 to 397 are the same as the title conclusions (are repeated). The abbreviation SOT appears in the abstract but has not been introduced previously.

Response to Comment 3, Reviewer 1:

Thank you for bringing these typographical errors to our attention. We have made the necessary revisions accordingly.

Reviewer 2 Report

Comments and Suggestions for Authors

In the current systematic review, the authors have reviewed the current knowledge of immunity against tetanus, diphtheria, and pertussis in adult SOT recipients where they concluded that those recipients experienced considerable immunity, but this immunity decreases over time. Overall, the review is well-written and I have some comments for the authors

1. In the abstract, write the full name of the abbreviation SOT in its first mention.

2. The current study focused on adult SOT recipients. It would be interesting if the authors compared the output of their analysis with similar studies that focused on children recipients. Can age affect immunity against targeted microorganisms in SOT recipients?

3. The last paragraph of the discussion and the conclusion are the same. Either delete the duplication or introduce your idea differentially.

Comments on the Quality of English Language

Minor editing of English language required

Author Response

Comments from Reviewer 2:

In the current systematic review, the authors have reviewed the current knowledge of immunity against tetanus, diphtheria, and pertussis in adult SOT recipients where they concluded that those recipients experienced considerable immunity, but this immunity decreases over time. Overall, the review is well-written and I have some comments for the authors

Comment 1, Reviewer 2:

  1. In the abstract, write the full name of the abbreviation SOT in its first mention.

Response to Comment 1, Reviewer 2:

Thank you for bringing this typographical error to our attention. We revised accordingly.

Comment 2, Reviewer 2:

  1. The current study focused on adult SOT recipients. It would be interesting if the authors compared the output of their analysis with similar studies that focused on children recipients. Can age affect immunity against targeted microorganisms in SOT recipients?

Response to Comment 2 Reviewer 2:

As you correctly pointed out, age can be one of the factors that can affect antibody response. We considered this issue and summarized data in sub-section 3.4, “Gender and age differences in antibody response. To highlight this, we elaborated in the discussion and added three new references. Please see, lines 348-367:

 “There is a noticeable decline in immunity to tetanus, diphtheria, and pertussis post-transplantation [10,18,21]. However, the decline in immunity is not uniform for all components of the Td/Tdap vaccine. A study in kidney transplant recipients revealed that while nearly all transplant recipients maintained protective levels of tetanus antibodies after a year, a significant portion (38%) had diphtheria antitoxin levels below the protective threshold [23]. This variation in immunity can be attributed to several factors. For instance, post-transplant immunosuppressive treatment can lead to reduced protective antibody levels [17,18]. A study from France indicated that lung transplant recipients receiving rituximab had a lower immune response to the tetanus vaccine than those not on rituximab [25]. Immunosuppressive agents that are used as maintenance therapy in SOT recipients, such as cyclosporine and tacrolimus, primarily affect T cell immunity. However, the combination therapy can impact both hu-moral and cellular immune responses to vaccination. This underscores the importance of studying the effects of different immunosuppressive regimens on both short-term and long-term responses to vaccination.”

It now reads:

“There is a noticeable decline in immunity to tetanus, diphtheria, and pertussis post-transplantation [10,18,21]. However, the decline in immunity is not uniform for all components of the Td/Tdap vaccine. A study in kidney transplant recipients revealed that while nearly all transplant recipients maintained protective levels of tetanus antibodies after a year, a significant portion (38%) had diphtheria antitoxin levels below the protective threshold [23]. This variation in immunity can be attributed to several factors. Gender and age have been investigated as factors influencing immunity; however, the findings in studies that we included were inconsistent [9, 18, 19, 23]. There appears to be a trend toward diminished immunity with age, which may be attributed to a reduction in immune function associated with aging, a process known as immunosenescence [27]. However, studies including pediatric SOT recipients have also shown a decline in immune response to diphtheria and tetanus vaccines [28, 29]. Thus, factors other than age may also play a significant role. For instance, post-transplant immunosuppressive treatment can lead to reduced protective antibody levels [17,18]. A study from France indicated that lung transplant recipients receiving rituximab had a lower immune response to the tetanus vaccine than those not on rituximab [25]. Immunosuppressive agents that are used as maintenance therapy in SOT recipients, such as cyclosporine and tacrolimus, primarily affect T cell immunity. However, the combination therapy can impact both hu-moral and cellular immune responses to vaccination. This underscores the importance of studying the effects of different immunosuppressive regimens on both short-term and long-term responses to vaccination.”

References:

  1. Pollard, A.J.; Bijker, E.M. A Guide to Vaccinology: From Basic Principles to New Developments. Nat. Rev. Immunol. 2021, 21, 83–100, doi:10.1038/s41577-020-00479-7.
  2. Enke, B.U.; Bökenkamp, A.; Offner, G.; Bartmann, P.; Brodehl, J. Response to Diphtheria and Tetanus Booster Vaccination in Pediatric Renal Transplant Recipients. Transplantation 1997, 64, 237–241, doi:10.1097/00007890-199707270-00010.
  3. Pedrazzi, C.; Ghio, L.; Balloni, A.; Panuccio, A.; Foti, M.; Edefonti, A.; Assael, B.M. Duration of Immunity to Diphtheria and Tetanus in Young Kidney Transplant Patients. Pediatr. Transplant. 1999, 3, 109–114, doi:10.1034/j.1399-3046.1999.00013.x.

Comment 3, Reviewer 2:

  1. The last paragraph of the discussion and the conclusion are the same. Either delete the duplication or introduce your idea differentially.

Response to Comment 1, Reviewer 2:

 We revised accordingly.

Reviewer 3 Report

Comments and Suggestions for Authors

The authors Emil Lenzing et al. in their manuscript describe a review of 11 papers about the difference of induced or natural immunization for the diseases tetanus, diphtheria and pertussis, in patients subject to transplantation.

The work is well structured.

I suggest the authors specify the three diseases better in the introduction, to give a sense of reading to new readers as well, and to explain well in the text how they excluded the 299 papers from the 315. They certainly used the tool described but it would be good if they detailed well in materials and methods how this bias tool works.

Otherwise, the work meets my expectations

Author Response

Comments from Reviewer 3:

The authors Emil Lenzing et al. in their manuscript describe a review of 11 papers about the difference of induced or natural immunization for the diseases tetanus, diphtheria and pertussis, in patients subject to transplantation. The work is well structured.

Comment 1, Reviewer 3:

I suggest the authors specify the three diseases better in the introduction, to give a sense of reading to new readers as well, and to explain well in the text how they excluded the 299 papers from the 315. They certainly used the tool described but it would be good if they detailed well in materials and methods how this bias tool works. Otherwise, the work meets my expectations

Response to Comment 1, Reviewer 3:

Thank you for your comments. As you and other respected reviewers suggested we elaborated on the introduction, please see on page 1, line 43, to page 2 line 52, which read:

” Tetanus, diphtheria, and pertussis are rare bacterial VPIs in adult SOT recipients [3–6]. In a recent cohort of SOT recipients from Switzerland, no case of diphtheria or tetanus was found within 12 years post-transplantation; however, two cases of pertussis were detected, resulting in an incidence rate of 10 per 100,000 person-years of follow-up [2]. Despite the low incidence of the infections, vaccination with tetanus and diphtheria toxoids vaccine (Td), or tetanus, diphtheria, and acellular pertussis (Tdap) vaccine are recommended in adult SOT recipients before or after transplantation [7,8]. The recommendations are mainly due to the potential risk of severe infection and poor outcomes; however, they primarily rely on studies conducted on children, and data on adult SOT recipients are scarce [7,8].”

It now reads:

” Tetanus, diphtheria, and pertussis are rare bacterial VPIs in adult SOT recipients [3–6]. In a recent cohort of SOT recipients from Switzerland, no case of diphtheria or tetanus was found within 12 years post-transplantation; however, two cases of pertussis were detected, resulting in an incidence rate of 10 per 100,000 person-years of follow-up [2]. Despite the low incidence of tetanus, diphtheria, and pertussis infections, treating these infections in SOT recipients is challenging and requires specialized intensive care units with high mortality [3]. A case report from Brazil documented a patient acquiring tetanus six years after renal transplantation, leading to acute kidney injury and a 37-day hospital stay [3]. In France, another case involved a renal transplant recipient who developed generalized tetanus 12 years post-transplantation; although anti-tetanus antibodies were detectable, the patient required prolonged intensive care and could not ingest food for 11 days due to trismus [4]. Additional reports from the USA and Spain have described pertussis infections in renal transplant recipients that were diagnosed and treated after significant delays, with patients experiencing a month of coughing [5, 6]. Due to the potential risk of severe infection and poor outcomes, vaccination with tetanus and diphtheria toxoids vaccine (Td), or tetanus, diphtheria, and acellular pertussis (Tdap) vaccine are recommended in adult SOT recipients before or after transplantation [7,8]. However, the recommendations primarily rely on studies conducted on children, and data on adult SOT recipients are scarce [7,8].

Moreover, we mentioned details about the risk of bias on page 3, lines 138-147 and page 4, lines 148-152.

We mentioned the screening process on page 2, lines 85-92 which read:

“Two independent investigators (EL and OR) performed the initial screening of papers, utilizing predetermined search terms and evaluating titles and abstracts for potential relevance. Subsequently, relevant papers were thoroughly assessed in full-text, and inclusion criteria were rigorously applied to determine their eligibility for the review. In instances where discrepancies arose during the screening and inclusion process, a third in-dependent investigator (ZBH) was involved to resolve any disagreements and ensure consistency. We also reviewed the reference list of the included studies and conducted a manual search on Google to find any relevant studies not identified through the mentioned databases.”

Furthermore, we elaborated about Covidence on page 3, line 121, which read:

“We extracted data using Extraction 2.0 in the Covidence online platform [13].”,

It now reads:

“We extracted data using Extraction 2.0 in the Covidence online platform [13]. Covidence is a web-based software that assists in the systematic review process through sever-al steps. A project is introduced to the platform, and investigators are invited via email. The investigators can import search results from different databases into the project and the platform helps to remove duplicate studies found across several databases. Each investigator can independently screen the imported studies by title and abstract; if a study seems relevant, they can include it for full-text review or exclude it, providing a reason. It is possible to design data extraction forms and extract data directly within the platform. Additionally, Covidence supports communication and task management among team members.”